# The Relationship between Teacher Leadership and School Climate: Findings from a Teacher-Leadership Project

Serigne Mbaye Gningue [1], Roger Peach [2], Adeeb M. Jarrah [1] and Yousef Wardat [3,*]

1   Emirates College for Advanced Education, Abu Dhabi P.O. Box 126662, United Arab Emirates
2   Lehman College of the City of New York (CUNY), New York, NY 10468, USA
3   Department of Curriculum and Instruction, Faculty of Mathematics Education, Higher College of Technology, Al Ain 17155, United Arab Emirates
*   Correspondence: ywardat1@hct.ac.ae

**Abstract:** A positive school climate and teacher leadership have both been shown to have beneficial effects on student achievement. This study was part of a wider research effort designed to assess the effects of a teacher-leadership development project. We hypothesized that there was a positive relationship between teacher leadership development and school climate. Seventy project participants from two cohorts responded to a teacher-leadership survey and 891 personnel from 42 schools from which participant teachers were drawn responded to a school climate survey. We found that, generally, there was little relationship between school climate and teacher-leadership development. However, a more fine-grained analysis showed that, for Cohort 2, schools that encourage teacher-to-teacher interactions are likely to see personal growth and development in teacher leaders in their staff. Additional findings suggest that if teacher-to-teacher interactions are encouraged, then teachers will increase their development as teacher leaders. However, as the results are correlational and not causal, it may be that, as teachers engage more in professional development activities, they encourage more positive teacher-to-teacher interactions in their school.

**Keywords:** teacher leadership; school climate; teacher-to-teacher interactions; Bronx; NY; United Arab Emirates (UAE)

## 1. Introduction

A positive school climate has been associated with several advantageous educational outcomes. For example, it can lead to an increase in students' academic success and achievement levels and a reduction in maladaptive behavior [1,2]; lead to an increase in job satisfaction for teachers and administrators [3]; and make the transition to a new school easier for students [4]. By contrast, a negative school climate can inhibit optimal learning and development [2,5–8].

In general, the focus of school-climate research has been on the school environment's relationship with students' academic achievement and welfare, and the role of the school principals in establishing a positive environment in the school [9–13]). The focus on the school principal's role in creating a positive school climate has tended to mean that, when research has attempted to examine the relationship between school climate and leadership, the principals' leadership style has been the focus of the research [10,14].

At about the same time, as there was a growing interest in the effect of school climate on students' academic achievement and welfare, theoreticians and researchers were advocating for a form of school-based leadership that was less hierarchical than one that mainly envisaged school leadership as residing in the school principal's office. This approach advocated for the use of experienced teachers as teacher-leaders in their schools and the wider educational community [15,16].

Teacher leadership has been increasingly associated with the practice of educational improvement [17–20]. Louis et al. [11] described leadership in terms of two core functions:

providing direction and exercising influence. These functions are enacted within particular contexts and have the potential to generate organizational reform. York-Barr and Duke [20] asserted that the concept of teacher leadership implies that teachers hold a central position in the ways schools operate and in the core functions of teaching and learning. In this way, teachers are given the power to help create a positive environment in their schools. Thus, teacher empowerment through taking on leadership roles becomes an important aspect of school climate [21–23].

## 1.1. Context of the Study

This study was part of a larger research endeavor that attempted to assess the impact of a teacher-leadership professional development project, the Mathematics Teacher Transformation Institutes (MTTI), on participants' classrooms practices, schoolwide culture, and student outcomes. MTTI was a Mathematics and Science Partnership (MSP) program funded by the National Science Foundation (NSF) aimed at building mathematics teacher leadership in Bronx middle and high schools working with mathematics teachers with at least four years' experience [5,24]. Bronx public schools serve a high proportion of low-income, Hispanic, and African-American students. This proportion is higher in the Bronx than any other borough in New York City. The Bronx continues to be the poorest borough in New York City. The United States Census Bureau [7] reported that the median household income between 2006 and 2010 was USD 34,264 in the Bronx compared with USD 55,603 for the whole of New York City. The latest Census Bureau (2021) reported that 27% of the Bronx population libr below the poverty line, about double the rate in New York City (13.6%), but more than double the rate in New York State (12.8%). There was a much-needed effort to develop experienced teachers as a resource for mathematics improvement at Bronx middle and high school levels.

MTTI focused on deepening participating teachers' mathematics-content knowledge, broadening their pedagogical repertoire through the process of inquiry, and developing their leadership capacities across a number of domains. It also strived to develop and sustain a professional community of teachers. The aim of the MTTI program was to develop informal rather than formal teacher leadership. That is to say, its general purpose was not to develop principals or assistant principals, although some participants might have attained such positions, but rather to develop teachers that work with and have influence on their school colleagues and the wider educational community to improve mathematics teaching in the Bronx and beyond [20,25]. MTTI was funded to support two cohorts of Bronx teachers (approximately 40 in each) over six years. Cohort 1 ended in June 2011 and Cohort 2 began right after and ended in 2014. Data collection continued through 2015–2016.

One problem with attempting to examine the relationship between teacher leadership and school climate is that there is a variety of conceptualizations of both concepts in the literature, which has led to the use of countless methodologies to define both constructs, making it difficult generalize findings [9,20,26]. However, our approach to teacher leadership and its development was based on a model postulated by Lord and Miller [27] that proposes that seven major types of leadership roles can be related to leadership development in mathematics. These seven types are as follows:

1. Type 1 (T1): In-classroom support of individual teachers—addressing the needs of individual teachers, feedback, modeling, team teaching;
2. Type 2 (T2): Professional development activities for groups of teachers—providing year-round workshops or institutes with follow-up in individual teachers' classrooms;
3. Type 3 (T3): Indirect support benefiting several classrooms—performing service on standards and curriculum committees;
4. Type 4 (T4): Crisis management—responding to the unexpected;
5. Type 5 (T5): Interactions with a larger educational community—networking with mathematics teachers from local schools, at the district level, or nationwide;
6. Type 6 (T6): Initiating extra-curricular mathematics activities—initiating a mathematics or robotic team or creating other extracurricular activities;

7.  Type 7 (T7): Initiating personal growth and professional development in mathematics—refining own teaching practices, classroom research.

To gather information on the school climate, we used the School Culture/Climate Survey (SCCS). The SCCS combines items from the teacher leadership survey developed for the Learning from Leadership research project (Louis et al [11]−Wallace Foundation) by the University of Minnesota and Wahlstrom and Louis (2008). The SCCS contains items pertaining to three aspects of school climate: school leadership style (whether distributed or hierarchical); teacher–teacher relationships, (such as speaking with colleagues about various instructional issues, observing each other's classrooms, and sharing lesson plans); and the school as a learning community. These three aspects combine to provide an overall school climate measure.

### 1.2. Purpose of Study

The main purpose of this study was to assess the relationship between teacher-leadership development and school climate. We examined the relationship between the seven types of teacher leadership, as postulated by Lord and Miller [27], and the three aspects of school climate proposed by Louis et al. [11]. We hypothesized that project participants would undertake teacher-leadership roles to a greater extent in schools with a more positive climate than in schools with a less positive climate.

### 1.3. Conceptual Framework

The roots of the MTTI theory of action can be traced from two conceptual directions. First, the theory of action relies on the intellectual development of the individual as a necessary prerequisite to change in culture. Empowering individuals with an increase in their expertise and intellectual capacity will build their personal self-esteem and ownership of the job, and lead them to seek opportunities to work cooperatively with colleagues and parents in ways that will help make schools a collaborative enterprise. Such an approach is partly analogous to cognitive theories of organizational learning [28,29] that view learning as being created via individuals' processing and transmission of information through communication, explanation, recombination, contrast, inference, and problem-solving [30].

Second, MTTI's approach also suggests that for individual change to collectively add up to cultural change, structures and processes are required to help define and shape the work of the collective on particular areas of the identified need. This notion is supported by theories of organizational learning that focus on the ways in which individuals learn in contexts, and the ways in which organizations themselves "learn" [31–33]. Additionally, Louis et al. [11] described leadership in terms of two core functions: providing direction and exercising influence. These functions, when enacted within particular contexts, have the potential to generate organizational reform. The definition that guides MTTI's conceptual framework is most closely aligned with the leadership-as-influence function, "a process by which teachers, individually or collectively, influence their colleagues to improve teaching and learning practices and outcomes" [20].

Thus, MTTI believed that increasing teachers' expertise and intellectual capacity through subtle but enduring professional development is a prerequisite to developing leadership capacity at the school and the potential for change. A teacher who is an excellent classroom instructor will lead by modeling the art of effective teaching. In this view, effective approaches to professional development are sustained over a period of from two to three years and immerse participants in multi-layered and scaffolder activities that focus on developing their content knowledge, pedagogical approaches, and leadership capacities, considering both the knowledge and expertise that participants bring to and from their classrooms and the daily contexts in which they work. The primary goal of this approach to professional development was to have a long-lasting impact on participants' practices—habits of mind and action—in their classrooms and within their schools [28–30].

Teacher empowerment will build personal self-esteem, ownership of the job, and personal interest in improving the performance of the organization. MTTI believes for

instance that teacher leaders empowered with the "cycle of inquiry" skills are more likely to engage their school in a culture of working together to identify and make decisions about school's progress and challenges. For individual change to collectively add up to cultural change, however, the enactment and establishment of certain structures and processes is required. Indeed, the teacher leader will be more likely to emerge and blossom in a system of shared leadership, in which groups of individuals interact, making decisions together and distributing roles, rather than in a system where the conceptions of leadership focus on the actions of singular individuals. MTTI believed that the role of the principal in this principle is critical. Therefore, how the school leadership is exercised in the school becomes crucial in creating opportunities for the teacher-leader to exercise leadership roles and responsibilities. As such opportunities are created, MTTI presumed that the possibility for teacher-to-teacher collaborations and dialogue with partners about improvement, as well as the chance to bring about changes in, or enhance school practices toward, a more distributed leadership.

## 2. Literature Review

Although both teacher leadership and a positive school climate have been seen to be important in developing student achievement, there has been relatively little research examining the relationship between these two to date. In one study, Sweetland and Hoy [34] examined the relationships between school climate, teacher empowerment, and student achievement. They found that in schools with collegial leadership and a high level of teacher professionalism, there was a high level of teacher empowerment that resulted in increased student achievement [22].

McCarley et al. [35] conducted a study that looked at the relationship between teacher assessments of a principal's transformational leadership qualities and the perceived school climate. A sample of 399 teachers from five high schools in a large urban school district in southeast Texas were given the Multifactor Leadership Questionnaire to assess their principal's transformational leadership and the Organizational Climate Description Questionnaire for Secondary Schools to assess the climate of their respective schools. The findings revealed a link between transformative leadership and supportive, engaged, and frustrated aspects of a school's atmosphere, indicating that quality leadership and a positive school atmosphere are essential to the success of every principal, student, and school.

Allen et al. [36] examined the relationship between transformational leadership, school climate, and student mathematics and reading achievements in a small suburban school district in southeast Texas. A purposive sample of elementary school principals were given the Multifactor Leadership Questionnaire (MLQ-5X) to assess how well a principal demonstrates the characteristics of a transformational leader based on teacher perceptions. In parallel, a convenience sample of teachers from the schools of those principals was surveyed using the School Climate Inventory-Revised (SCI-R) instrument. The findings revealed a link between transformational leadership and excellent school climate. However, neither transformative leadership nor school atmosphere were found to be associated with student achievement.

Dutta and Sahney [37] studied the relationships between the dimensions of principals' instructional and transformational leadership behaviors, teachers' perceptions of the school climate (social, affective, and physical environment), job satisfaction, and student achievement. Cross-sectional survey data from 306 secondary school principals and 1539 teachers of two regions in India revealed that principal leadership behaviors were not directly linked to teacher job satisfaction or school-wide student achievement. Rather, through the social and affective component of the school atmosphere, transformational leader behavior had an indirect effect on teacher job satisfaction. The physical environment, on the other hand, proved to have a significant role in mediating the effects of instructional leadership on teacher job satisfaction. Principals appear to prefer the former strategy when comparing the respective indirect effect sizes of instructional and transformational leadership behaviors on student achievement.

Kılınç [38] examined the relationships between primary school teachers' perceptions of school climate and teacher leadership. Using the Organizational Climate Description Questionnaire-RE and the Teacher Leadership Scale on 259 primary school teachers in India, they found that restrictive school atmosphere and teacher leadership had unfavorable and substantial connections. Restriction was also found to be a negative and significant predictor of all three subscales of teacher leadership (institutional improvement, professional improvement, and collaboration among colleagues). On the other hand, based on institutional progress, a directive school atmosphere was the sole positive and significant predictor of teacher leadership.

Bual and Madrigal [39] assessed the degree of school climate and extent of teacher leadership in Catholic schools in Antique, Philippines. They polled 486 administrators, teachers, and students using an adopted school climate questionnaire and a standardized teacher leadership measure. The descriptive–comparative and correlational study designs revealed that the school climate was very satisfactory, with relationship as the highest priority and physical resources as the lowest priority. Teacher leadership was widely practiced, with community as the highest priority and policy and professional learning as the lowest priority. A link between teacher leadership and the age, sex, work status, educational achievement, and professional standing of the instructors was found, and an association between school climate and teacher leadership was determined.

## 3. Methods

### 3.1. Teacher Leadership

3.1.1. Participants Selection

MTTI was funded to support two cohorts of 40 teachers with at least four years teaching experience over five years. The first cohort completed the program after three years in June 2011. The second cohort began immediately after and also lasted three years. Both cohorts took a set of 12 credits in mathematics content and 12 credits of education-based credits, mainly in action research and leadership. For both cohorts, there was no bias in the selection of teachers and schools, as participants were selected based on specific criteria. The recruitment call required a minimum of two teachers per school and a principal recommendation for interested Bronx teachers. Applicants were interviewed by the program leadership team with selection criteria that included background information (NYS Certification in Mathematics; status as a Bronx Middle or High School teacher; a master's degree; and a minimum 4 years of teaching experience) and a statement of interest about the program. Applicants who did not meet these requirements were not selected.

3.1.2. Teacher Leadership Survey (TLS)

A 40-item Teacher Leadership Survey (TLS), mostly taken from the instrument developed [11], was designed to determine various aspects of teacher-leadership development (TLD). The TLS combined items from the teacher leadership survey developed for the Learning from the leadership research project ([11]–Wallace Foundation) by the University of Minnesota and the PRISM Teacher leader Program, an NSF-funded program (2010) that defined and presented teacher leadership roles. MTTI research team (MTTI_RT) met with the PRIZM research team at the annual NSF conference on teacher leadership (2010) to discuss the different roles. The MTTI_RT then met with a group of experienced and retired mathematics teachers who were hired as consultants and assigned as mentors of MTTI participants. The group discussed the roles that mathematics teachers could possibly play in their schools, as members of school improvement committees, mentors, instructional specialists, catalysts for change, classroom supporters, resource providers, data coaches, learning facilitators, workshop leaders, conference organizers, modelers, and as users and providers of classroom technology. These interactions led to a consensus around the items to be included in the survey as teacher leadership roles.

Using a 6-point scale from None (1) to A Great Deal (6), the survey questions asked participants to determine the extent to which they practiced the different types of leadership

roles and responsibilities. The 40 items were classified under the types (T1, T2, T3, T5, T6, and T7) of mathematics leadership activities identified by Lord and Miller [40] and National Council of Supervisors of Mathematics [41]. No questions on the survey were related to crisis management (T4).

The TLS was administered to both cohorts at the beginning of the leadership component of the MTTI project. Thirty-one Cohort 1 participants and 39 participants from Cohort 2 completed the survey [42].

We constructed an overall teacher-leadership score by summing and then averaging ratings across the 40 items for each participant. Mean ratings were then calculated for each participant for the six types of teacher leadership roles and activities identified earlier. This was achieved by summing responses across items within each leadership type and then dividing this by the number of items in that leadership type to obtain a mean value for each respondent for each leadership type.

### 3.2. School Climate

For Cohort 1, the SCCS was distributed to teachers and administrators of participating MTTI schools in June 2010. For Cohort 2, the SCCS was administered to participants' schools in May 2012. We asked MTTI participants to distribute and collect the SCCS from as many personnel in their school as they could. For Cohort 1, 280 individuals from 16 of 32 participating schools returned the SCCS. As the response rate was quite low for Cohort 1, with only half the number of participating schools returning surveys, for Cohort 2, we offered small incentives (e.g., math manipulatives) to participating schools for completion and return of the SCCS. For Cohort 2, 983 SCCS surveys were distributed and 620 (63%) returned from 29 of 33 participating schools. Of these, nine were either returned blank, or had more than 15 consecutive answers rated in the same way. These nine were excluded from analysis, leaving a total of 611 (62%) of analyzable returns. It would appear from the increased number of responses for Cohort 2 that giving incentives for the return of the SCCS was effective.

Of the schools in both cohorts that returned the SCCS, three (two large high schools and one international school) had participants in both Cohort 1 and Cohort 2. Thus, 42 different schools were represented across both cohorts. The schools generally served a low-income Hispanic and African American community. On average, Cohort 1 schools had 65.2% (SD = 26.2%) of students receiving free lunches, with 56.1% (SD = 17.0%) of the student body Hispanic students, with a further 34.7% (SD = 14.2%) being African American students. Similarly, Cohort2 schools had, on average, 76.3% (SD = 21.9%) of students receiving free lunches, with 63.8% (SD = 10.4%) of the student body being Hispanic students, with a further 28.5% (SD = 11.6%) being African American students.

The SCCS in Table A1 in the Appendix A asks school staff who responded 36 Likert-scale statements or questions, such as "How much direct influence do school teams have on school decisions?" and "How many teachers in the school feel responsible to help each other improve their instruction?" Respondents were asked to rate all 36 items on the SCCS on a six-point scale (1 = not at all, 6 = to a great extent) regarding the extent to which they agreed with the statement or question.

From the 36 items on the modified SCCS, three separate indices were created: one of school leadership style (SLS) (10 items, (Qs. 2, 25–33)); one of teacher-to-teacher interactions (TTI) (9 items, (Qs. 3–7, 12–14 and 35)) and one of school as a learning community (SLC) (8 items, (Qs. 8–11, 15–18)). We also created an "overall school climate" score by averaging across all 27 item scores. The survey also included questions about respondents' beliefs about teacher leadership and their activity as a dean or an assistant principal. This paper focuses only on responses to the 27 school climate items.

## 4. Results

### 4.1. Teacher Leadership

Thirty-one (31) MTTI Cohort 1 and 39 Cohort 2 participants completed the TLS. None of the 40 items in the survey referred to crisis management (Type 4). We constructed an overall teacher-leadership score by summing and then averaging the 40 items. The overall mean leadership rating for Cohort 1 was 2.72 (SD = 1.0) out of a possible 6. The overall mean leadership rating for Cohort 2 was 2.52 (SD = 0.7). The mean scores for the six types of leadership for both cohorts are given in Table 1.

**Table 1.** Mean ratings of leadership activity by leadership type for Cohorts 1 and 2.

| | Cohort 1 | | | Cohort 2 | | | |
|---|---|---|---|---|---|---|---|
| Leadership Type | N | Mean | SD | N | Mean | SD | Cohen d |
| In-classroom support of individual teachers (T1). | 31 | 3.23 | 1.3 | 39 | 2.75 | 1.0 | 0.41 |
| Prof. development for groups of teachers (T2). | 31 | 3.16 | 1.2 | 39 | 2.61 | 0.9 | 0.52 |
| Indirect support for several classrooms (T3). | 31 | 2.37 | 0.9 | 39 | 2.37 | 0.8 | 0.0 |
| Interactions with the ed. community (T5). | 31 | 2.18 | 0.9 | 39 | 1.72 | 0.6 | 0.60 |
| Extra-curricular math activities (T6). | 31 | 2.31 | 1.4 | 39 | 1.73 | 1.0 | 0.48 |
| Personal growth and prof. development (T7). | 31 | 3.47 | 1.2 | 39 | 3.91 | 1.2 | 0.37 |

We compared the groups using Cohen's d, an effect size calculation that accounts for large differences in variance between the groups and differences in sample size. For this measure, a d value of 0.00–0.24 indicates no/negligible difference between the groups, whereas a value of 0.25–0.5 indicates a moderate to medium difference, and a value of 0.75–1.0 indicates a very large difference [43].

Both Cohort 1 and Cohort 2 teachers were most likely to have responsibility for teacher-leadership activities that fall within Type 1 (in-classroom support of individual teachers), Type 2 (professional development activities for groups of teachers) and Type 7 (initiating personal growth and professional development in mathematics) at the beginning of the MTTI program. Cohort 1 scored higher on Type 1 (d = 0.41) and Type 2 (d = 0.52) while Cohort 2 scored slightly higher on Type 7 (d = 0.37). Both cohort teachers were least likely to have responsibility for Type 5 (interactions with a larger educational community), and Type 6 (initiating extra-curricular mathematics activities), while practicing Type 3 (indirect support benefiting several classrooms) at the same level (mean = 2.37 for both cohorts, d = 0.0).

### 4.2. School Climate

For the SCCS, the overall mean and standard deviation (SD) for each item of both cohorts are given in Table A1 in the Appendix A. Overall, there were no significant differences between the mean ratings of Cohort 1 and Cohort 2 schools for almost all 36 items in the survey. Only Question 25 seemed to indicate a large difference between the cohorts (d = 0.52).

School Leadership Style (SLS). School leadership style items (Qs. 2, 25–33; Table 2) were intended to measure the degree to which leadership in the school was distributed across the faculty, as opposed to being concentrated in the administration. Examples of these items are: "Teachers have an effective role in school-wide decision making." and "The administration in my school establishes a climate that reinforces teachers' leadership activities."

Both Cohort 1 and Cohort 2 respondents gave the highest ratings in this category to question 30, asking whether their administration allowed teachers access to computerized school records' information, with Cohort 2 scoring slightly higher than Cohort 1. Cohort 1 assigned lowest ratings to question 25: "The department chairs/grade-level team leaders influence how money is spent in this school," while Cohort 2 assigned the lowest ratings to question 26: "teachers have an effective role in school-wide decision-making." The

overall average rating across all items in the category was 3.2 for Cohort 1 (SD = 0.8), and 3.6 (SD = 1.0) for Cohort 2. Except for question 25 where Cohort 2 scored higher (d = 0.52), the two cohorts showed minimal differences in most items of the SLS.

**Table 2.** School Leadership Style (SLS): means and Standard Deviations for Cohorts 1 and 2.

| | School Leadership Style (SLS) 10 Items (Qs. 2, 25–33) | Cohort 1 | | | Cohort 2 | | | |
|---|---|---|---|---|---|---|---|---|
| Q# | On a scale of 1 (none) -6 (a great deal), please indicate the level of each of the following | *n* | Mean | SD | *n* | Mean | SD | Cohen d |
| 2 | How much direct influence do school teams (depts., grade levels, other teacher groups) have on school decisions? | 276 | 3.43 | 1.2 | 595 | 3.51 | 1.3 | 0.06 |
| 25 | The department chairs/grade-level team leaders influence how money is spent in this school. | 276 | 2.68 | 1.5 | 594 | 3.44 | 1.4 | 0.52 |
| 26 | Teachers have an effective role in school-wide decision-making. | 277 | 2.95 | 1.4 | 603 | 3.18 | 1.3 | 0.17 |
| 27 | Teachers have a significant input into plans for professional development and growth. | 278 | 3.06 | 1.5 | 604 | 3.31 | 1.5 | 0.17 |
| 28 | School principal(s) ensures wide participation in decisions about school improvement. | 272 | 3.18 | 1.5 | 607 | 3.41 | 1.4 | 0.16 |
| 29 | The administration in my school allows teachers released time to perform leadership tasks | 267 | 2.95 | 1.5 | 601 | 3.4 | 1.4 | 0.31 |
| 30 | The administration in my school allows teachers access to computerized information that is required for various analyses | 267 | 4.07 | 1.6 | 601 | 4.62 | 1.5 | 0.36 |
| 31 | The administration in my school establishes a climate that reinforces teachers' leadership activities | 267 | 3.27 | 1.5 | 603 | 3.63 | 1.4 | 0.25 |
| 32 | The administration in my school supports the creation and/or continuation of extra-curricular mathematics activities | 265 | 3.27 | 1.5 | 577 | 3.81 | 1.5 | 0.36 |
| 33 | The administration in my school supports the offering of advanced placements courses | 261 | 3.28 | 1.8 | 589 | 3.88 | 1.6 | 0.35 |
| | Overall Mean | 280 | 3.2 | 0.8 | 611 | 3.6 | 1.0 | 0.44 |

Teacher-to-Teacher Interactions (TTI). Questions in this category (Qs. 3–7, 11–14 and 35; Table 3) concerned the ways in which teachers in the school support each other, regarding the curriculum, instruction, school rules, and in other ways. Examples of these items include: "How often have you visited other teachers' classroom to observe instruction?" and "How often have you exchanged suggestions for curriculum materials with colleagues?"

Both Cohort 1 and Cohort 2 gave their highest ratings to Q7, "How often in this school year have you had conversations with colleagues about what helps students learn best?" and Q6, about the frequency with which teachers discussed managing classroom behavior among themselves. The least frequent interaction was having a colleague visit one's class (Q12). The average rating across all schools was 3.69 (SD = 0.9) for Cohort 1 and 3.7 (SD = 1.0) for Cohort 2. Both cohorts responded almost equally to all questions.

School as Learning Community (SLC): The remaining school climate questions (Qs. 8–10, 15–18; Table 4) measured various aspects of the school climate concerning shared values and goals and working together. Items include, for instance, "Teachers support the principal in enforcing school rules" and question 17, "In our school we have well-defined learning expectations for all students."

**Table 3.** Teacher-To-Teacher Interactions (TTI)-(Qs. 3–7, 12–14 and 35): Means and Standard Deviations for Cohorts 1 and 2.

| Q# | Teacher-To-Teacher Interactions (TTI) 9 Items, (Qs. 3–7, 12–14 and 35) On a scale of 1 (none) -6 (a great deal), please indicate the level of each of the following | Cohort 1 | | | Cohort 2 | | | Cohen d |
|---|---|---|---|---|---|---|---|---|
| | | *n* | Mean | SD | *n* | Mean | SD | |
| 3 | How often in this school year have you exchanged suggestions for curriculum materials with colleagues? | 276 | 4.19 | 1.5 | 608 | 4.1 | 1.5 | 0.06 |
| 4 | How often in this school year have you had conversations with colleagues about the goals of this school? | 272 | 4.12 | 1.4 | 609 | 4.06 | 1.5 | 0.04 |
| 5 | How often in this school year have you had conversations with colleagues about development of new curriculum? | 268 | 3.97 | 1.5 | 608 | 3.82 | 1.4 | 0.10 |
| 6 | How often in this school year have you had conversations with colleagues about managing classroom behavior? | 272 | 4.46 | 1.4 | 508 | 4.44 | 1.4 | 0.01 |
| 7 | How often in this school year have you had conversations with colleagues about what helps students learn best? | 272 | 4.48 | 1.3 | 609 | 4.44 | 1.4 | 0.03 |
| 12 | How often in this school year have you had colleagues observe your classroom? | 273 | 2.69 | 1.6 | 587 | 2.79 | 1.5 | 0.06 |
| 13 | How often in this school year have you received meaningful feedback on your performance from colleagues? | 274 | 3.1 | 1.5 | 593 | 3.17 | 1.4 | 0.05 |
| 14 | How often in this school year have you visited other teachers' classrooms to observe instruction? | 277 | 2.91 | 1.6 | 599 | 2.98 | 1.5 | 0.05 |
| 35 | I help other teachers deal with classroom management | 266 | 3.31 | 1.7 | 599 | 3.57 | 1.8 | 0.15 |
| | Overall | 280 | 3.69 | 0.9 | 611 | 3.7 | 1.0 | 0.01 |

**Table 4.** School As A Learning Community (SLC)-[Qs. 8–11, 15–18]. Means and Standard Deviations for Cohorts 1 and 2.

| Q# | School as a Learning Community (SLC) 8 items, (Qs. 8–11, 15–18) On a scale of 1 (none) -6 (a great deal), please indicate the level of each of the following | Cohort 1 | | | Cohort 2 | | | Cohen d |
|---|---|---|---|---|---|---|---|---|
| | | *n* | Mean | Std | *n* | Mean | Std | |
| 8 | How many teachers in this school feel responsible to help each other improve their instruction? | 268 | 3.82 | 1.4 | 604 | 3.81 | 1.6 | 0.01 |
| 9 | How many teachers in this school take responsibility for improving the school outside their own class? | 276 | 3.55 | 1.3 | 606 | 3.54 | 1.3 | 0.01 |
| 10 | How many teachers in this school help maintain discipline in the entire school, not just their classroom? | 271 | 3.46 | 1.3 | 608 | 3.61 | 1.4 | 0.11 |
| 11 | How often in this school year have you invited someone in to help teach your class(es)? | 273 | 1.96 | 1.3 | 587 | 2.37 | 1.3 | 0.32 |
| 15 | Teachers support the principal in enforcing school rules. | 267 | 3.98 | 1.5 | 599 | 4.19 | 1.4 | 0.14 |
| 16 | Most teachers in our school share a similar set of values, beliefs, and attitudes related to teaching and learning. | 267 | 4.02 | 1.3 | 607 | 4.03 | 1.3 | 0.01 |
| 17 | In our school we have well-defined learning expectations for all students. | 265 | 3.93 | 1.4 | 610 | 4.09 | 1.5 | 0.11 |
| 18 | Our student assessment practices reflect our curriculum standards. | 263 | 4.07 | 1.3 | 606 | 4.35 | 1.4 | 0.21 |
| | Overall Mean | 280 | 3.6 | 0.8 | 611 | 3.8 | 1.0 | 0.22 |

For both Cohort 1 and Cohort 2, the highest ratings were assigned to question 18: "Our student assessment practices reflect our curriculum standards." Question 16 "most teachers in our school share a similar set of values, beliefs, and attitudes related to teaching and learning" was also highly and equally rated between the two cohorts (mean= 4.02 vs. 4.03, d = 0.01). Similarly, the lowest ratings for both cohorts were question 11, "How often in this school year have you invited someone in to help teach your class(es)?" (mean = 1.96 vs. 2.37; d= 0.32). The average rating across all variables was 3.6 (SD = 0.8) for all Cohort 1 schools, and 3.8 (SD = 1.0) for all Cohort 2 schools. Except for question 11, rating differences were negligible for almost all SLC questions, with d ranging from 0.01 to 0.22.

Overall, School Climate. Cohort 1 results showed that, for all three sub-scales, school leadership style (SLS), teacher-to-teacher interactions (TTI), and school as a learning community (SLC), averages across all schools were in the 3–4 points range, out of a maximum of 6. Responses to all 27 school climate items were also summed and averaged to create an overall measure of school climate. This summary variable ranged from 3.3 to 4.3 for each Cohort 1 school; that is, most school faculties gave their schools "mid-range" ratings on the three above dimensions of school culture. There were no significant differences between schools on this variable.

### 4.3. Relationships among School Climate and Teacher-Leadership Activities

Cohort 1. Overall measures of the three aspects of school culture were regressed on the overall rating of MTTI participants' teacher-leadership activities, and then on each of the six sub-categories of teacher-leadership activities separately at the 0.05 level. Two significant relationships were found. "School as Learning Community" (SLC) significantly predicted two of the types of TL activities: "in-classroom support of individual teachers" and" indirect support benefiting several classrooms." Variance in SLC scores accounted for 19.8% of the variation in ratings for the in-classroom support of individual teachers, and 32.4% of the variance in the indirect support benefiting several classrooms. Tables 5 and 6 below show that, on average, for every one-point increase in ratings for SLC, ratings for in-classroom support of individual teachers increased by almost two points (*B* = 1.88), and indirect support benefiting several classrooms increased by 1.5 points (*B* = 1.48).

**Table 5.** School as Learning Community by in-classroom support.

| | Unstandardized Coefficients | | | |
| Model | B | Std. Error | t | Sig. |
|---|---|---|---|---|
| 1 School as Learning Com. | 1.879 | 0.869 | 2.163 | 0.044 |

**Table 6.** School as Learning Community by Indirect Support to Groups of Teachers.

| | Unstandardized Coefficients | | | |
| Model | B | Std. Error | t | Sig. |
|---|---|---|---|---|
| 1 School as Learning Com. | 1.480 | 0.491 | 3.016 | 0.007 |

"Overall school culture" was very close to significantly predicting indirect support benefiting several classrooms (Table 7). Variations in overall school culture ratings accounted for 15.3% of the variation in indirect support, benefiting several classrooms.

**Table 7.** School Culture Overall by Indirect Support to Groups of Teachers.

| | Unstandardized Coefficients | | | |
| Model | B | Std. Error | t | Sig. |
|---|---|---|---|---|
| 1 School Culture Overall | 0.861 | 0.420 | 2.049 | 0.055 |

Cohort 2: For Cohort 2, the only significant relationships, as follows: initiating personal growth and professional development in mathematics was positively correlated with both school leadership style (r = 0.603 ($p$ = 0.002)) and teacher-to-teacher interaction (r = 0.614 ($p$ = 0.002)). Teacher-to-teacher interaction was also positively correlated with professional development activities for groups of teachers (r = 0.479 ($p$ = 0.021)).

Teacher Leadership Activities Electronic Logs (e-Logs) and Consultant Reports

While these Cohort 1 findings were promising in providing some support for the affirmation that the extent of collaborative culture in the school contributes to the degree to which MTTI teachers provide leadership for other teachers in their schools, we took caution in interpreting the results. Some schools had only two or three responses, thus providing less precise estimates of their schools' population mean than other schools with a greater number of responses. Three MTTI schools were not represented in the survey: two MTTI teachers had changed schools mid-year, and another was not able to provide the data. Some schools provided very few responses. We compared MTTI participants' responses to the survey, firstly to those of non-MTTI teachers, and then non-MTTI mathematics teachers. There was no significant difference in either case, pushing us to be more cautious. The number of MTTI participants who either responded to or identified themselves as MTTI teachers in the survey was small (six in all). We decided to make it optional for teachers to identify themselves and their subjects, since some schools have only a few teachers in a subject area. It would have been easy to recognize them as responders. This issue was raised by Cohort 1 MTTI participants, who explained that many teachers did not want to be identified, especially when responding about leadership issues. Some MTTI teachers identified themselves, but only a few. Moreover, the sample of mathematics teachers in most schools was very small, making the disaggregation by subject area almost meaningless. Some schools had 2–3 mathematics teachers and could count up to 30 different subject area teachers. Since Cohort 2 schools were similar in nature, the research team decided to collect more qualitative data with Cohort 2, in the form of case studies, "Teacher Leadership E-log" and monthly "Consultant Reports", to obtain a more complete record of the various teacher leadership activities that teachers engage in and determine whether participants' responses to the teacher-leadership survey (TLS) and teachers' responses to the climate survey would be corroborated by these qualitative data. We asked MTTI teachers to complete the "Teacher Leadership E-log" at least once a month, starting in October 2011. The e-log form asks about activities involving one-on-one teacher interactions, work with groups of teachers, activities that indirectly benefit teachers in the school, and other types of leadership activities.

We analyzed the first 141 responses of 37 Cohort 2 teachers (90% of the cohort) between October 2011 and April 2012 to refine and better classify the areas of reporting. For instance, teachers reported relatively more (100) instances of working with groups of teachers, and slightly fewer (70) working with other teachers one-on-one. There were 49 reports involving "other" leadership activities, which we could not define at the time. From reading the entire 141 logs though, it was obvious that some teachers may have made mistakes in how they classified their activities. We recoded "other activities" into one of the other two categories. The final analysis comprised 340 logs received between October 2011 and May 2013, with most teachers submitting between 5 and 14 reports. Responses in the area of "Work with Groups of Teachers," were more evenly distributed across the various categories, and no one sort of activity was the most common. "Working . . . to solve mathematical problems" and "Organizing . . . a professional learning community . . . " were the most frequent activities, followed by "providing resources . . . . to other teachers" and "Engaging in discussion of multiple paths to solutions of problems."

The section of the log that asked teachers to report activities that "indirectly benefit teachers in your school" (Type 3), listed a variety of such activities. The most frequently mentioned in this category were work on standards and curriculum committees, especially with the Common Core Learning Standards (CCLS), and work with community groups. Teacher reports of CCLS-related activities (Type 3) rose from 10 mentions among 81 reports

in the first term of Cohort II's participation in MTTI to approximately one activity for every three reports filed in the following spring and fall; these then fell off slightly in the spring of 2013. This finding seems to validate the results obtained in Tables 6 and 7.

As for the Consultant Reports, 29 MTTI teachers were visited in their schools by a MTTI teacher-consultant (TC) during the spring 2013 semester. Each teacher was visited once a month over that period of time. During each visit, the TC asked the MTTI participant to describe any teacher-leadership activities with which they had been involved. The log asked about the teacher's individual goals and how they had been working to achieve them. It also asked about the leadership activities they had been engaged in. MTTI teachers were also asked about the focus or recipient of the activities (e.g., colleagues, administration, students, parents or others). One question asked about the leadership categories to which the activity belonged (leadership of self, leadership of colleagues, or leadership in the wider community). Another inquired about the primary area of teacher practice that was addressed (e.g., equity, teaching and learning, curriculum, or assessment). Finally, teachers were asked whether the school climate helped or hindered their development as a teacher leader.

A range of activities from 21 of the participants showed that teachers mainly mentored their colleagues and, quite often, this involved working with them in relation to the introduction of the math Common Core Learning Standards. They also worked with them on developing curriculum and assessment methods (Type 3). Two are involved in preparing Math Olympiad teams, and two others run out-of-school-hours math seminars for their students. There are also reports of various conference attendances.

## 5. Discussion

The results of this research were obtained across two cohorts of teachers and their schools. A total of 42 separate schools provided responses to the SCCS. These schools had a high percentage of Hispanic and African American students, and a large percentage of students who received free lunch. This suggests that the findings are based on a reasonably representative sample of Bronx schools, although it could be argued that schools with a positive climate are more likely to encourage their teachers to participate in a leadership development program, and thus the sample might be less representative of all Bronx middle and high schools.

There was little relationship between school climate and teacher leadership for both cohorts, and the relationship differed across cohorts. Despite this overall lack of relationship between school climate and teacher leadership, there were some results that could have an impact on the development of both teacher leadership and school climate. For example, findings from Cohort 2 suggest the schools that encourage teacher-to-teacher interaction are likely to see personal growth and development as teacher leaders in their staff (Kilinç, 2014 [38]; Bual & Madrigal, 2021 [39]). However, as the results are correlational and not causal, it may be that as teachers engage more in professional development activities, they encourage more positive teacher-to-teacher interactions in their school.

In terms of school leadership style, results from both cohorts indicate that teachers are usually given a reasonable amount of access to computerized records and information. This might be in response to the demands of the Department of Education for administrators and teachers to use a more data-driven approach to learning. At the same time, teachers seem to have a more limited input into school-wide decision-making, even if these decisions impact their professional development and growth.

These findings suggest that teachers, quite properly, are mainly engaged in conversations aimed at helping students learn. However, there is less opportunity to observe one another's teaching and develop their pedagogy and that of their colleagues in this way. This may be due to the way the teaching day is scheduled, meaning that some teachers are teaching their individual classes at the same time as others.

As a learning community, a school's assessment practices generally reflect its curriculum standards, and most teachers in the school share a similar set of values, beliefs, and

attitudes related to teaching and learning. It is perhaps to be expected that assessment standards would reflect curriculum standards, and that a reasonably cohesive school would have staff that share a similar set of values and attitudes to teaching and learning. However, taken together with the teacher-to-teacher interactions results, it seems as if teaching is still a relatively 'lonely' occupation, with limited opportunity for teachers to visit one another's classrooms and learn from each other's pedagogy [44].

## 6. Conclusions

With 280 faculty responses from 16 Cohort-1-participating schools and 623 faculty responses from 29 Cohort-2-participating schools, about 22% of all Bronx middle and high schools were represented in this study. These schools had a high percentage of Hispanic and African American students, and a large percentage of students who received free lunch. This suggests that the findings are based on a reasonably representative sample of Bronx schools, although it could be argued that schools with a positive climate are more likely to encourage their teachers to participate in a leadership development program, and thus the sample might be less representative of all Bronx middle and high schools.

On average, both cohorts rated the overall school climate at about 3.5 on a six-point scale. This suggests that most schools who provided teachers for the MTTI project had a school climate that was not seen as particularly supportive of teacher development, nor was it particularly inhibiting. The principals of these schools indicated that they would support the teachers in the MTTI program in their leadership activities. This might indicate that schools with teachers in the MTTI program had a more positive school climate than other equivalent schools in the Bronx. Indeed, principals' interview data in the final year of the MTTI program show that the impact on teachers' practice and on schools might have been greater than was reflected in the survey.

## 7. Limitations of the Study

This study was based on MTTI school participants' self-report of the degree to which they viewed leadership roles and interactions in their schools. Questioning the validity of survey data is often one of the first reactions when survey results are shared [45]. To increase the validity of the assessment of teacher-leadership activities, the MTTI research team obtained data from direct observations of leadership activities and asked MTTI teacher-consultants (another facet of the program) to collect leadership information from participants at monthly interviews. However, we believed that the principals insights could represent a strong source of knowledge about teacher leadership development and its link to school climate. However, it was difficult to involve them on a regular basis.

## 8. Recommendations

The study revealed that there was little link between school atmosphere and teacher leadership development in general. More detailed research revealed that schools that foster teacher-to-teacher engagement in Cohort 2 are more likely to experience personal growth and development as teacher leaders in their workforce. Additional data imply that encouraging teacher-to-teacher contact would help to instructors develop as teacher leaders.

In their responses, principals focused on the "big picture," a macro-perception that things were better and that the MTTI teachers were leaders who contributed to the improvement of their school. The micro-details, such as specific approaches to classroom instruction that may have contributed to that improvement, appeared to be less valued and ignored.

Using the survey for the entire school rather than just for the mathematics department or clusters faculty might have contributed to lowering the average of all items pertaining to three aspects of school climate: school leadership style (whether distributed or hierarchical); teacher–teacher relationships, (such as speaking with colleagues about various instructional issues, observing each other's classrooms, and sharing lesson plans); the school as a learning community and, subsequently, the overall school climate measure (Louis et al., 2010).

Mathematics teachers are most likely to be involved in teacher-to-teacher interactions with fellow mathematics teachers in the same department or cluster, and with other teachers as well. Future studies could look at such relationships within clusters and departments rather than within the entire school.

**Author Contributions:** Conceptualization, Y.W. and R.P.; methodology, A.M.J. and S.M.G.; software, Y.W., A.M.J. and S.M.G.; validation, R.P., A.M.J., Y.W., and S.M.G.; formal analysis, Y.W. and S.M.G.; investigation, Y.W.; resources, Y.W., S.M.G., R.P. and A.M.J.; data curation, Y.W. and A.M.J.; writing—original draft preparation, Y.W.; writing—review and editing, Y.W., S.M.G., R.P. and A.M.J.; visualization, Y.W.; supervision, R.P., A.M.J., Y.W., and S.M.G.; project administration, Y.W. and R.P.; funding acquisition, A.M.J. All authors have read and agreed to the published version of the manuscript.

**Funding:** This research was the result of funding from the National Science Foundation (NSF) Award #DUE-0832247. Mathematics Teacher Transformation Institutes (MTTI) at Lehman College of the City University of New York (CUNY).

**Institutional Review Board Statement:** The study was conducted in accordance with the Declaration of Helsinki and approved by the Ethics Committee of City University of New York (CUNY). Protocol code ERS_0832247 Serigne M. Gningue was a faculty at Lehman College, MTTI Co-PI and head of the research team.

**Informed Consent Statement:** Informed consent was obtained from all subjects involved in the Study.

**Data Availability Statement:** Not applicable.

**Conflicts of Interest:** The authors declare no conflict of interest.

## Appendix A

**Table A1.** School Climate/Culture Survey (SCCS)—Means and Standard Deviations for Individual Items for Cohorts 1 and 2.

| | On a Scale of 1 (None) -6 (A Great Deal); Please Indicate the Level of Each of the Following: | Cohort 1 | | | Cohort 2 | | | |
|---|---|---|---|---|---|---|---|---|
| | | *n* | Mean | SD | *n* | Mean | SD | Cohen d |
| 1 | How much direct influence do students have on school decisions? | 276 | 2.95 | 1.40 | 604 | 2.89 | 1.60 | 0.04 |
| 2 | How much direct influence do school teams (depts., grade levels, other teacher groups) have on school decisions? | 276 | 3.43 | 1.20 | 595 | 3.51 | 1.30 | 0.06 |
| 3 | How often in this school year have you exchanged suggestions for curriculum materials with colleagues? | 276 | 4.19 | 1.50 | 608 | 4.10 | 1.50 | 0.06 |
| 4 | How often in this school year have you had conversations with colleagues about the goals of this school? | 272 | 4.12 | 1.40 | 609 | 4.06 | 1.50 | 0.04 |
| 5 | How often in this school year have you had conversations with colleagues about development of new curriculum? | 268 | 3.97 | 1.50 | 608 | 3.82 | 1.40 | 0.10 |
| 6 | How often in this school year have you had conversations with colleagues about managing classroom behavior? | 272 | 4.46 | 1.40 | 508 | 4.44 | 1.40 | 0.01 |
| 7 | How often in this school year have you had conversations with colleagues about what helps students learn best? | 272 | 4.48 | 1.30 | 609 | 4.44 | 1.40 | 0.03 |
| 8 | How many teachers in this school feel responsible to help each other improve their instruction? | 268 | 3.82 | 1.40 | 604 | 3.81 | 1.60 | 0.01 |
| 9 | How many teachers in this school take responsibility for improving the school outside their own class? | 276 | 3.55 | 1.30 | 606 | 3.54 | 1.30 | 0.01 |
| 10 | How many teachers in this school help maintain discipline in the entire school, not just their classroom? | 271 | 3.46 | 1.30 | 608 | 3.61 | 1.40 | 0.11 |
| 11 | How often in this school year have you invited someone in to help teach your class(es)? | 273 | 1.96 | 1.30 | 587 | 2.37 | 1.30 | 0.32 |
| 12 | How often in this school year have you had colleagues observe your classroom? | 274 | 2.69 | 1.60 | 593 | 2.79 | 1.50 | 0.06 |

**Table A1.** *Cont.*

| | On a Scale of 1 (None) -6 (A Great Deal); Please Indicate the Level of Each of the Following: | Cohort 1 | | | Cohort 2 | | | |
|---|---|---|---|---|---|---|---|---|
| | | *n* | Mean | SD | *n* | Mean | SD | Cohen d |
| 13 | How often in this school year have you received meaningful feedback on your performance from colleagues? | 277 | 3.10 | 1.50 | 599 | 3.17 | 1.40 | 0.05 |
| 14 | How often in this school year have you visited other teachers' classrooms to observe instruction? | 274 | 2.91 | 1.60 | 604 | 2.98 | 1.50 | 0.05 |
| 15 | Teachers support the principal in enforcing school rules. | 267 | 3.98 | 1.50 | 599 | 4.19 | 1.40 | 0.14 |
| 16 | Most teachers in our school share a similar set of values, beliefs, and attitudes related to teaching and learning. | 267 | 4.02 | 1.30 | 607 | 4.03 | 1.30 | 0.09 |
| 17 | In our school we have well-defined learning expectations for all students. | 265 | 3.93 | 1.40 | 610 | 4.09 | 1.50 | 0.11 |
| 18 | Our student assessment practices reflect our curriculum standards. | 263 | 4.07 | 1.30 | 606 | 4.35 | 1.40 | 0.21 |
| 19 | Generally speaking, teachers' mastery of academic content contributes to their role as a teacher leader in their school. | 276 | 4.47 | 1.40 | 604 | 4.91 | 1.40 | 0.31 |
| 20 | A teacher leader's influence is exerted primarily in the classroom. | 276 | 4.03 | 1.40 | 598 | 4.08 | 1.50 | 0.03 |
| 21 | A teacher leader's influence is exerted primarily in the content area department. | 276 | 3.94 | 1.40 | 593 | 4.19 | 1.40 | 0.18 |
| 22 | A teacher leader's influence is exerted primarily in the school community. | 272 | 3.63 | 1.50 | 593 | 4.10 | 1.40 | 0.32 |
| 23 | A teacher leader's influence is exerted primarily in the neighborhood community. | 268 | 2.65 | 1.50 | 589 | 3.15 | 1.60 | 0.32 |
| 24 | Teacher leaders tend to emerge by their own actions and knowledge rather than being assigned to that role by the principal. | 272 | 4.01 | 1.50 | 600 | 4.49 | 1.40 | 0.33 |
| 25 | The department chairs/grade-level team leaders influence how money is spent in this school. | 276 | 2.68 | 1.50 | 594 | 3.44 | 1.40 | 0.52 |
| 26 | Teachers have an effective role in school-wide decision-making. | 277 | 2.95 | 1.40 | 603 | 3.18 | 1.30 | 0.17 |
| 27 | Teachers have a significant input into plans for professional development and growth. | 278 | 3.06 | 1.50 | 604 | 3.31 | 1.50 | 0.17 |
| 28 | School principal(s) ensures wide participation in decisions about school improvement. | 272 | 3.18 | 1.50 | 607 | 3.41 | 1.40 | 0.16 |
| 29 | The administration in my school allows teachers released time to perform leadership tasks | 267 | 2.95 | 1.50 | 601 | 3.40 | 1.40 | 0.31 |
| 30 | The administration in my school allows teachers access to computerized information that is required for various analyses | 267 | 4.07 | 1.60 | 601 | 4.62 | 1.50 | 0.36 |
| 31 | The administration in my school establishes a climate that reinforces teachers' leadership activities | 267 | 3.27 | 1.50 | 603 | 3.63 | 1.40 | 0.25 |
| 32 | The administration in my school supports the creation and/or continuation of extra-curricular mathematics activities | 265 | 3.27 | 1.50 | 577 | 3.81 | 1.50 | 0.36 |
| 33 | The administration in my school supports the offering of advanced placements courses | 261 | 3.28 | 1.80 | 589 | 3.88 | 1.60 | 0.35 |
| 34 | I sometimes act as a dean | 266 | 2.59 | 1.80 | 598 | 2.71 | 1.90 | 0.06 |
| 35 | I help other teachers deal with classroom management | 266 | 3.31 | 1.70 | 599 | 3.57 | 1.80 | 0.15 |
| 36 | I have written reports about other teachers' performance in place of the AP | 250 | 1.30 | 1.00 | 578 | 1.74 | 1.20 | 0.40 |

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
