# Peer review of "The Relationship between Teacher Leadership and School Climate: Findings from a Teacher-Leadership Project"

_education, doi:10.3390/educsci12110749_

Round 1
Reviewer 1 Report
Overall, the article is fairly written, despite a few things being noted for revision.
Repetition in describing the purpose of the study. Please check the purpose of the study and the conceptual framework.
s.d or SD for standard deviation? Please check.
The study’s limitations should appear at the end of the discussion of the results.
I suggest adding references from the latest sources, especially in the discussion section.
Author Response
|
Repetition in describing the purpose of the study. Please check the purpose of the study and the conceptual framework.
|
Repetition deleted from conceptual framework-Lines 174-179 |
|
SD for standard deviation |
Std was defined for Standard Deviation (Std) on line 324.
26 replacements were made changing SD and s.d. to Std |
|
The study’s limitations should appear at the end of the discussion of the results. |
Limitations of the study was added from Line 592 after conclusions. |
|
Suggests adding references from the latest sources, especially in the discussion section |
|

Reviewer 2 Report
Thank you for the opportunity to read this interesting manuscript; it addresses what are clearly important policy issues. Suggestions to improve the manuscript are detailed below.
Lines 15-16: The wording here suggests a causal relationship between encouraging teacher-to-teacher interactions and enhanced teacher leadership, but the data are cross-sectional and thus do not lend themselves to strong causal assertions (especially since few - if any - controls were added to the models [and no random assignment of teachers/schools]).
Lines 17-18: These keywords do not seem relevant to this manuscript.
Lines 543-588: The qualitative findings with respect to principals' assessments comes across as a nearly separate study. Could the authors juxtapose key findings from the interviews with the survey data in just 2-4 sentences and provide a citation to this other part of the broader study? In my opinion, the authors do not sufficiently help the reader to "square" these disparate quantitative and qualitative findings from the vantage points of the different actors (e.g., which analysis should carry more weight?). As they stand, the inclusion of these lines may serve to obfuscate the manuscript's findings and recommendations.
The authors carried out dozens of hypothesis tests (including the single items). With an alpha of 0.05, it would be expected that they would commit a Type I error once every 20 tests. Could the lion's share of their significant findings simply be attributable to testing error?
How were schools and teachers in the Bronx selected to participate in the MTTI? Could there be sample selection bias that affects the manuscript's findings? For instance, if schools that made it into the sample generally boasted some of the most favorable climates (and their principals and participating teachers were already among the most open to teacher leadership development), might these selection factors make it more difficult to detect strong relationships between climate and leadership development?
A potential source of error could stem from the fact that schools' climates were reported by teachers of many subjects (not just math-related) and principals, but the leadership types were reported by math teachers only (and with at least 4 years of experience). In other words, it would seem more appropriate to model math teachers' perceptions of school climate on leadership types.
How much of a time lag was there between data collection on climate and leadership for each cohort? If time lags were sizable, could this bias the relationship too weak between the two constructs? A related question: how stable should climate measurements be year-over-year?
Cronbach's alphas should be reported and interpreted for all indexes (or composite reliabilities). Do these indexes have acceptable validity, at least as reported by extant studies involving other populations?
Is there a reason (theoretical or practical) to expect that Cohorts 1 and 2 would differ on leadership types? Why not run an independent samples t-test to see if they differ significantly on leadership? If not, I suggest combining the two cohorts to increase statistical power for the later regressions/correlations involving climate and teacher leadership. When carrying out the regressions/correlations, the authors could then test for any conditional effects of climate on leadership.
In addition to p values, effect sizes would be helpful to report. The bivariate regression coefficients were not reported in tabular format; were a considerable number of the coefficients of at least moderate strength but simply not at conventional levels of significance? By the way, which alpha level was used to conduct significance testing? Since the authors strongly hypothesized positive relationships between climate and leadership development, they might consider reporting one-tailed tests instead.
Tables 1 and 2 should be combined for "side-by-side" direct comparison between cohorts (including p values and Cohen's d effect sizes).
Given their lengths, perhaps Tables 3 and 4 could be placed in appendices. Finally, given the amount of repetition in Tables 3 and 4, perhaps they could be combined to offer a side-by-side comparison of the cohorts.
The section on "School Climate" beginning at Line 322 could be shortened considerably. The textual reporting on Tables 3 and 4 separately by cohort is repetitive.
Author Response
|
Reviewer 2 |
|
|
Lines 15-16: The wording here suggests a causal relationship between encouraging teacher-to-teacher interactions and enhanced teacher leadership, but the data are cross-sectional and thus do not lend themselves to strong causal assertions (especially since few - if any - controls were added to the models [and no random assignment of teachers/schools]). |
The following was added: “However, as the results are correlational and not causal, it may be that as teachers engage more in professional development activities, they encourage more positive teacher-to-teacher interactions in their school.” |
|
Lines 17-18: These keywords do not seem relevant to this manuscript. |
Agreed. Key words were changed. |
|
Lines 543-588: The qualitative findings with respect to principals' assessments comes across as a nearly separate study. Could the authors juxtapose key findings from the interviews with the survey data in just 2-4 sentences and provide a citation to this other part of the broader study? In my opinion, the authors do not sufficiently help the reader to "square" these disparate quantitative and qualitative findings from the vantage points of the different actors (e.g., which analysis should carry more weight?). As they stand, the inclusion of these lines may serve to obfuscate the manuscript's findings and recommendations. |
Lines 553-588 were summarized and shortened as suggested. Please see Lines 559-567. |
|
The authors carried out dozens of hypothesis tests (including the single items). With an alpha of 0.05, it would be expected that they would commit a Type I error once every 20 tests. Could the lion's share of their significant findings simply be attributable to testing error?
How were schools and teachers in the Bronx selected to participate in the MTTI? Could there be sample selection bias that affects the manuscript's findings? For instance, if schools that made it into the sample generally boasted some of the most favourable climates (and their principals and participating teachers were already among the most open to teacher leadership development), might these selection factors make it more difficult to detect strong relationships between climate and leadership development?
A potential source of error could stem from the fact that schools' climates were reported by teachers of many subjects (not just math-related) and principals, but the leadership types were reported by math teachers only (and with at least 4 years of experience). In other words, it would seem more appropriate to model math teachers' perceptions of school climate on leadership types.
How much of a time lag was there between data collection on climate and leadership foreach cohort? If time lags were sizable, could this bias the relationship too weak between the two constructs? A related question: how stable should climate measurements be year-over-year?
Cronbach's alphas should be reported and interpreted for all indexes (or composite reliabilities). Do these indexes have acceptable validity, at least as reported by extant studies involving other populations?
Is there a reason (theoretical or practical) to expect that Cohorts 1 and 2 would differ on leadership types? Why not run an independent samples t-test to see if they differ significantly on leadership? If not, I suggest combining the two cohorts to increase statistical power for the later regressions/correlations involving climate and teacher leadership. When carrying out the regressions/correlations, the authors could then test for any conditional effects of climate on leadership.
In addition to p values, effect sizes would be helpful to report. The bivariate regression coefficients were not reported in tabular format; were a considerable number of the coefficients of at least moderate strength but simply not at conventional levels of significance? By the way, which alpha level was used to conduct significance testing?
|
A section for Participants selection was added from Line 242-255 |
|
Since the authors strongly hypothesized positive relationships between climate and leadership development, they might consider reporting one-tailed tests instead.
Tables 1 and 2 should be combined for "side-by-side" direct comparison between cohorts (including p values and Cohen's d effect sizes).
Given their lengths, perhaps Tables 3 and 4 could be placed in appendices. Finally, given the amount of repetition in Tables 3 and 4, perhaps they could be combined to offer a side-by-side comparison of the cohorts.
The section on "School Climate" beginning at Line 322 could be shortened considerably. The textual reporting on Tables 3 and 4 separately by cohort is repetitive. |
Tables 1 and 2 were consolidated Tables 3 and 4 were also consolidated Narrative describing results were revised accordingly. Lines 378-530 |

Reviewer 3 Report
The article is focused on an interesting topic, because the cooperation of teachers and linking the content of the subject is a problem of education in many countries. The approach of school principals to this issue is also different. Perhaps it would be interesting to focus another study on a comparison with the issue of another country.
Reviewer 4 Report
Title: The Relationship between Teacher Leadership and School Climate: Findings from a Teacher-Leadership Project
The main purpose of this study was to evaluate the relationship between teacher development and its relationship with school climate.
The paper has an acceptable structure and scientific writing, but also has some weaknesses. Lines of professional development with organizational transformation intersect. This field of research is interesting.
However, there are weaknesses that are difficult to resolve:
- There are no perceptible ethical issues in the completion of the scales by the respondents.
- The study seems to have been conducted more than 10 years ago, which calls into question its relative value and the relevance of its publication: "For Cohort 1, the SCCS was distributed to teachers and administrators of participating MTTI schools in June 2010. For Cohort 2, the SCCS was administered to participants' schools in May 2012"
- The interpretation of the results based essentially on the average value of the responses obtained for each item of the likert scale seems insufficient to me.
- The use of data from interviews with principals in the conclusion of the results is incomprehensible, when these interviews are not mentioned in the methodology, nor in the interpretation of the results.
- The results are unreliable for establishing any relationship between school environment, leadership, and teacher professional development.
Although the authors dominate the knowledge of the subject, as revealed in the literature review, the work is little innovative and the results are poorly consolidated.
Author Response
|
Reviewer 4 |
As a reader, the only demand I have for the authors is that they explain the process by which they select the items that make up the questionnaire by following the one proposed by Louis et al. (2010). |
Please see additions in Lines 245-258. |
|
I welcome the inclusion of a recommendations section based on the results of the study. |

Reviewer 5 Report
The article presents a quantitative research on the identification of work environments in which teacher-leadership is promoted.
The article is well structured. The introduction adequately justifies the research, which is revealed as an area still in need of further study. the theoretical framework and literature review are adequate and very informative for the reader.
The methodology chosen is quantitative and well justified. As a reader, the only demand I have for the authors is that they explain the process by which they select the items that make up the questionnaire by following the one proposed by Louis et al. (2010). I believe that this aspect would make a relevant methodological design aspect transparent.
From here, the results clearly derive in the conclusions presented, which are relevant to the educational community. I welcome the inclusion of a recommendations section based on the results of the study.
Round 2
Reviewer 4 Report
See "teach-er-to-teacher", line 18.
I think they should address the ethics of insvetigation.
Author Response
Dear reviewer,
I have addressed teach-er-to-teacher", line 18.
Regards